# Achieving Equalized Odds by Resampling Sensitive Attributes

**Yaniv Romano**
Department of Statistics
Stanford University
Stanford, CA, USA
yromano@stanford.edu

**Stephen Bates**
Department of Statistics
Stanford University
Stanford, CA, USA
stephenbates@stanford.edu

**Emmanuel J. Candès**
Departments of Mathematics
and of Statistics
Stanford University
Stanford, CA, USA
candes@stanford.edu

## Abstract

We present a flexible framework for learning predictive models that approximately satisfy the equalized odds notion of fairness. This is achieved by introducing a general discrepancy functional that rigorously quantifies violations of this criterion. This differentiable functional is used as a penalty driving the model parameters towards equalized odds. To rigorously evaluate fitted models, we develop a formal hypothesis test to detect whether a prediction rule violates this property, the first such test in the literature. Both the model fitting and hypothesis testing leverage a resampled version of the sensitive attribute obeying equalized odds, by construction. We demonstrate the applicability and validity of the proposed framework both in regression and multi-class classification problems, reporting improved performance over state-of-the-art methods. Lastly, we show how to incorporate techniques for equitable uncertainty quantification—unbiased for each group under study—to communicate the results of the data analysis in exact terms.

## 1   Introduction

Machine learning algorithms are now frequently used to inform high-stakes decisions—and even to make them outright. As such, society has become increasingly critical of the ethical implications of automated decision making, and researchers in algorithmic fairness are responding with new tools. While fairness is context dependent and may mean different things to different people, a suite of recent work has given rise to a useful vocabulary for discussing fairness in automated systems [1, 2, 3, 4, 5, 6, 7]. Fairness constraints can often be articulated as conditional independence relations, and in this work we will focus on the *equalized odds* criterion [8], defined as

$$\hat{Y} \perp\!\!\!\perp A \mid Y, \tag{1}$$

where the relationship above applies to test points; here, $Y$ is the response variable, $A$ is a sensitive attribute (e.g. gender), $X$ is a vector of features that may also contain $A$, and $\hat{Y} = \hat{f}(X)$ is the prediction obtained with a *fixed* prediction rule $\hat{f}(\cdot)$. While the idea that a prediction rule obeying the equalized odds property is desirable has gained traction, actually finding such a rule for a real-valued or multi-class response is a relatively open problem. Indeed, there are only a few recent works attempting this task [9, 10]. Moreover, there are no existing methods to rigorously check whether a learned model achieves this property.

We address these two questions by introducing a novel training scheme to fit models that approximately satisfy the equalized odds criterion and a hypothesis test to detect when a prediction rule violates this same criterion. Both solutions build off of one key idea: we create a synthetic version

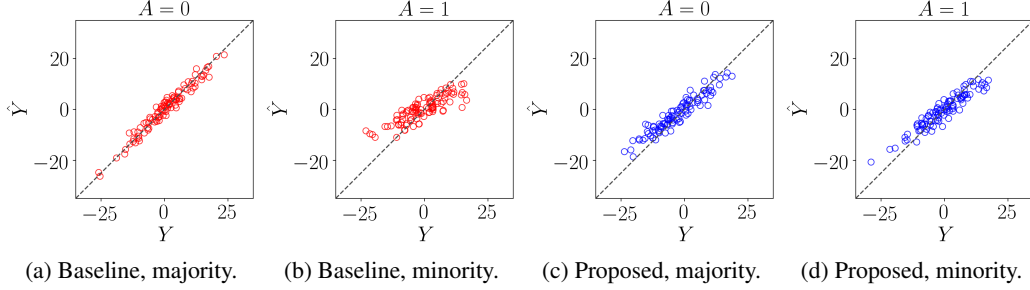

| (a) Baseline, majority. | (b) Baseline, minority. | (c) Proposed, majority. | (d) Proposed, minority. |

Figure 1: The effect of our learning framework on simulated data: (a,b) predictions from the baseline linear model; (c,d) predictions from the linear model fitted with the proposed equalized odds penalty.

$\tilde{A}$ of the sensitive attribute such that the triple $(\hat{Y}, \tilde{A}, Y)$ obeys (1) with $\tilde{A}$ in lieu of $A$. To achieve equitable model fits, we regularize our models toward the distribution of the synthetic data. Similarly, to test whether equalized odds holds, we compare the observed data to a collection of artificial data sets. The synthetic data is straightforward to sample, making our framework both simple to implement and modular in that it works together with any loss function, architecture, training algorithm, and so on. Based on real data experiments on both regression and multi-class classification tasks, we find improved performance compared to state-of-the-art methods.

## 1.1 A synthetic example

To set the stage for our methodology, we first present an experiment demonstrating the challenges of making equitable predictions as well as a preview of our method's results. We simulate a regression data set with a binary sensitive attribute and two features:

$$(X_1, X_2) \mid (A = 0) \overset{d}{=} (Z_1, 3Z_2) \quad \text{and} \quad (X_1, X_2) \mid (A = 1) \overset{d}{=} (3Z_1, Z_2),$$

where $Z_1, Z_2 \sim \mathcal{N}(0, 1)$ is a pair of independent standard normal variables, and the symbol $\overset{d}{=}$ denotes equality in distribution. We create a population where 90% of the observations are from the group $A = 0$ in order to investigate a setting with a large majority group. After conditioning on $A$, the model for $Y \mid X$ is linear: $Y = X^\top \beta_A + \epsilon$, with noise $\epsilon \sim \mathcal{N}(0, 1)$ and coefficients $\beta_0 = (0, 3)$ and $\beta_1 = (3, 0)$. We designed the model in this way so that the distribution of $Y$ given $X$ is the same for the two groups, up to a permutation of the coordinates. (In some settings, we might say that both groups are therefore equally deserving.) Consequently, the best model has equal performance in both groups. We therefore find it reasonable to search for a fitted model that achieves equalized odds in this setting.

To serve as an initial point of comparison, we first fit a classic linear regression model with coefficients $\hat{\beta} \in \mathbb{R}^2$ on the training data, minimizing the mean squared error. Figures (1a) and (1b) show the performance of the fitted model for each group on a separate test set. The fitted model performs significantly better on the samples from the majority group $A = 0$ than those from the minority group $A = 1$. This is not surprising since the model seeks to minimize the overall prediction error. Here, the overall root mean squared error (RMSE) evaluated on test points is equal to 2.40, with an average value of 1.79 for group $A = 0$ and of 5.48 for group $A = 1$. It is visually clear that for any vertical slice of the graph at a fixed value of $Y$, the distribution of $\hat{Y}$ is different in the two classes, i.e. the equalized odds property in (1) is violated. This fact can be checked formally with our hypothesis test for (1) described later in Section 3. The resulting p-value on the test set is 0.001 providing rigorous evidence that equalized odds is violated in this case.

Next, we apply our proposed fitting method (see Section 2) on this data set. Rather than a naive least squares fit, we instead fit a linear regression model that approximately satisfies the equalized odds criterion. The new predictions are displayed in Figures (1c) and (1d). In contrast to the naive fit, the new predictive model achieves a more balanced performance across the two groups: the blue points are dispersed similarly in these two panels. This observation is consistent with the results of our hypothesis test; the p-value on the test set is equal to 0.476, which provides no indication that the equalized odds property is violated. Turning to the statistical efficiency, the equitable model has improved performance for observations in the minority group $A = 1$ with an RMSE equal to 3.31, at

the price of reduced performance in the majority group $A = 0$, where the RMSE rises to 3.41. The overall RMSE is 3.40, larger than that of the baseline model.

## 1.2 Related work

The notion of equalized odds as a criterion for algorithmic fairness was introduced in [8]. In the special case of a binary target variables and a binary response variable, the aforementioned work offered a procedure to post-process any predictive model to construct a new model achieving equalized odds, possibly at the cost of reduced accuracy. Building off this notion of fairness, [11] and [12] show how to fit linear and kernel classifiers that are aligned with this criterion as well—these methods apply when the response and sensitive attribute are both binary. Similarly, building on the Hirschfeld-Gebelein-Renyi (HGR) Maximum Correlation Coefficient, [10] introduces a penalization scheme to fit neural networks that approximately obey equalized odds, applying to continuous targets and sensitive attributes. Coming at the problem from a different angle, [13, 9] fit models with an equalized odds penalty using an adversarial learning scheme. The main idea behind this method is to maximize the prediction accuracy while minimizing the adversary's ability to predict the sensitive attribute. Our method has the same objective as the latter two, but uses a new subsampling technique for regularization, which also leads to the first formal test of the equalized odds property in the literature.

## 2 Fitting fair models

### 2.1 Regularization with fair dummies

This section presents a method for fitting a predictive function $\hat{f}(\cdot)$ on i.i.d. training data $\{(X_i, A_i, Y_i)\}$ indexed by $i \in \mathcal{I}_{\text{train}}$ that approximately satisfies the equalized odds property (1). In regression settings, let $\hat{Y} = \hat{f}(X) \in \mathbb{R}$ be the predicted value of the continuous response $Y \in \mathbb{R}$. In multi-class classification problems where the response variable $Y \in \{1, \ldots, L\}$ is discrete, we take the output of the classifier to be $\hat{Y} = \hat{f}(X) \in \mathbb{R}^L$, a vector whose entries are estimated probabilities that an observation with $X = x$ belongs to class $Y = y$. We use this formulation of $\hat{Y}$ because it is the typical information available to the user when deploying a neural network for regression or classification, and our methods will use neural networks as the underlying predictive model. Nonetheless, the material in this subsection holds for any formulation of $\hat{Y}$, such as an estimated class label.

Our procedure starts by constructing a *fair dummy* sensitive attribute $\tilde{A}_i$ for each training sample:

$$\tilde{A}_i \sim P_{A|Y}(A_i \mid Y_i), \quad i \in \mathcal{I}_{\text{train}},$$

where $P_{A|Y}$ denotes the conditional distribution of $A_i$ given $Y_i$. This sampling is straightforward; see (4) below. Importantly, we generate $\tilde{A}_i$ without looking at $\hat{Y}_i$ so that we have the following property:

$$\hat{Y}_i \perp\!\!\!\perp \tilde{A}_i \mid Y_i, \quad i \in \mathcal{I}_{\text{train}}. \tag{2}$$

Notice that the above is exactly the equalized odds relation in (1), with a crucial difference that the original sensitive attribute $A_i$ is replaced by the artificial one $\tilde{A}_i$. We will leverage this fair, synthetic data for both model fitting and hypothesis testing in the remainder of this work.

Motivated by (2), we propose the following objective function for equalized odds model fitting:

$$\hat{f}(x) = \underset{f \in \mathcal{F}}{\operatorname{argmin}} \frac{1-\lambda}{|\mathcal{I}_{\text{train}}|} \sum_{i \in \mathcal{I}_{\text{train}}} \ell(Y_i, f(X_i)) + \lambda \mathcal{D}\left((\hat{\mathbf{Y}}, \mathbf{A}, \mathbf{Y}), (\hat{\mathbf{Y}}, \tilde{\mathbf{A}}, \mathbf{Y})\right). \tag{3}$$

Here, $\ell(\cdot)$ is a loss function that measures the prediction error, such as the mean squared error for regression, or the cross-entropy for multi-class classification. The second term on the right hand side is a penalty promoting the equalized odds property, described in detail soon. The hyperparameter $\lambda$ trades off accuracy versus equalized odds. Above, the $i$th row of $\hat{\mathbf{Y}} \in \mathbb{R}^{|\mathcal{I}_{\text{train}}| \times k}$ is $f(X_i) \in \mathbb{R}^k$, with $k = 1$ in regression and $k = L$ in multi-class classification. Similarly, we define $\mathbf{X} \in \mathbb{R}^{|\mathcal{I}_{\text{train}}| \times p}$ $\mathbf{A} \in \mathbb{R}^{|\mathcal{I}_{\text{train}}|}$, $\tilde{\mathbf{A}} \in \mathbb{R}^{|\mathcal{I}_{\text{train}}|}$, and $\mathbf{Y} \in \mathbb{R}^{|\mathcal{I}_{\text{train}}|}$, whose entries correspond to the features, sensitive attributes, fair dummies, and labels, respectively. As a result, both $(\hat{\mathbf{Y}}, \mathbf{A}, \mathbf{Y})$ and $(\hat{\mathbf{Y}}, \tilde{\mathbf{A}}, \mathbf{Y})$ are

matrices of size $|\mathcal{I}_{\text{train}}| \times (k+2)$. The function $\mathcal{D}(\mathbf{U}, \mathbf{V})$ is any measure of the discrepancy between two probability distributions $P_U$ and $P_V$ based on the samples $\mathbf{U}$ and $\mathbf{V}$, summarizing the differences between the two samples into a real-valued score. A large value suggests that $P_U$ and $P_V$ are distinct, whereas a small value suggests that they are similar. We give a concrete choice based on adversarial classification in Section 2.2. Since $(\hat{\mathbf{Y}}, \tilde{\mathbf{A}}, \mathbf{Y})$ obeys the equalized odds property by construction, making the discrepancy with $(\hat{\mathbf{Y}}, \mathbf{A}, \mathbf{Y})$ small forces the latter to approximately obey equalized odds.

**Proposition 1.** *Take $(X, A, Y) \sim P_{XAY}$ and set $\hat{Y} = \hat{f}(X)$ for some fixed $\hat{f}(\cdot)$ (again, $X$ may include $A$). Let $\tilde{A}$ be sampled indpendently from $P_{A|Y}(A|Y)$.[1] Then, $\hat{Y} \perp\!\!\!\perp A \mid Y$ if and only if $(\hat{Y}, A, Y) \stackrel{d}{=} (\hat{Y}, \tilde{A}, Y)$.*

The proof of this proposition as well as all other proofs are in Supplementary Section S1. We argue that this equivalence is particularly fruitful: indeed, if we find a prediction rule $\hat{f}(\cdot)$ such that $(\hat{\mathbf{Y}}, \mathbf{A}, \mathbf{Y})$ has the same distribution as $(\hat{\mathbf{Y}}, \tilde{\mathbf{A}}, \mathbf{Y})$ (treating the prediction rule as fixed), then $\hat{f}(\cdot)$ exactly satisfies equalized odds. Motivated by this, our penalty drives the model to a point where these two distributions are close based on the training set. When this happens, then, informally speaking, we expect that equalized odds approximately holds for future observations.

The regularization term in (3) can be used with essentially any existing machine learning framework, allowing us to fit a predictive model that is aligned with the equalized odds criterion, no matter whether the response is discrete, continuous, or multivariate. It remains to formulate an effective discriminator $\mathcal{D}(\cdot)$ to capture the difference between the two distributions, which we turn to next.

## 2.2  The discrepancy measure

A good discrepancy measure $\mathcal{D}(\cdot)$ should detect differences in distribution between the training data and the fair dummies in order to better promote equalized odds. Many examples have already been developed for the purpose of two-sample tests; examples include the Friedman-Rafsky test [14], the popular maximum mean discrepancy (MMD) [15], the energy test [16], and classifier two-sample tests [17, 18]. The latter are tightly connected to the idea of generative adversarial networks [19] which serves as the foundation of our procedure.

To motivate our proposal, suppose we are given two independent data sets $\{U_i\}$ and $\{V_i\}$: the first contains samples of the form $U_i = (\hat{Y}_i, A_i, Y_i)$, and the second includes $V_i = (\hat{Y}_i, \tilde{A}_i, Y_i)$. Our goal is to design a function that can distinguish between the two sets, so we assign a positive (resp. negative) label to each $U_i$ (resp. $V_i$) and fit a binary classifier $\hat{d}(\cdot)$. Under the null hypothesis that $P_U = P_V$, the classification accuracy of $\hat{d}(\cdot)$ on hold-out points should be close to $1/2$, while larger values provide evidence against the null. To turn this idea into a training scheme, we repeat the following two steps: first, we fit a classifier $\hat{d}(\cdot)$ whose goal is to recognize any difference in distribution between $U$ and $V$, and second, we fit a prediction function $\hat{f}(\cdot)$ that attempts to "fool" the classifier $\hat{d}(\cdot)$ while also minimizing the prediction error. In our experiment, the function $\hat{d}(\cdot)$ is formulated as a deep neural network with a differentiable loss function, so as the two models—$\hat{f}(\cdot)$ and $\hat{d}(\cdot)$—can be simultaneously trained via stochastic gradient descent.

While adversarial training is powerful, it can be sensitive to the choice of parameters and requires delicate tuning [13, 9]. To improve stability, we add an additional penalty that forces the relevant second moments of $U$ and $V$ to approximately match; we penalize by $\|\text{cov}(\hat{\mathbf{Y}}, \mathbf{A}) - \text{cov}(\hat{\mathbf{Y}}, \tilde{\mathbf{A}})\|^2$ where $\tilde{\mathbf{A}}$ is as in (2) and cov denotes the covariance, since under equalized odds this would be zero in the population (because $(\hat{Y}, A) \stackrel{d}{=} (\hat{Y}, \tilde{A})$ by Proposition 1). Combining all of the above elements, we can now give the full proposed procedure in Algorithm 1.

## 2.3  Sampling fair dummies

To apply the proposed framework we must sample fair dummies $\tilde{A}$ from the distribution $P_{A|Y}$. Since this distribution is typically unknown, we use the training examples $\{(A_i, Y_i)\}_{i \in \mathcal{I}_{\text{train}}}$ to estimate the

**Algorithm 1** Fair Dummies Model Fitting

---

**Input**: Data $\{(X_i, A_i, Y_i)\}_{i \in \mathcal{I}_{\text{train}}}$; predictive model $\hat{f}_{\theta_f}(\cdot)$ and discriminator $\hat{d}_{\theta_d}(\cdot)$.

1: **for** $k = 1, \ldots, K$ **do**
2:     Sample fair dummies $\tilde{A}_i \sim P_{A|Y}(A_i \mid Y_i), i \in \mathcal{I}_{\text{train}}$. See Section 2.3 for details.
3:     Update the discriminator parameters $\theta_d$ by repeating the following for $N_g$ gradient steps:

$$\mathcal{J}_d(\theta_d) = \frac{-1}{|\mathcal{I}_{\text{train}}|} \sum_{i \in \mathcal{I}_{\text{train}}} \left[ \log\left( \hat{d}_{\theta_d}\left( \hat{f}_{\theta_f}(X_i), A_i, Y_i \right) \right) + \log\left( 1 - \hat{d}_{\theta_d}\left( \hat{f}_{\theta_f}(X_i), \tilde{A}_i, Y_i \right) \right) \right]$$

$$\theta_d \leftarrow \theta_d - \mu \nabla_{\theta_d} \mathcal{J}_d(\theta_d)$$

4:     Update the predictive model parameters $\theta_f$ by repeating the following for $N_g$ gradient steps:

$$\mathcal{J}_f(\theta_f) = \frac{1 - \lambda}{|\mathcal{I}_{\text{train}}|} \sum_{i \in \mathcal{I}_{\text{train}}} \ell\left( Y_i, \hat{f}_{\theta_f}(X_i) \right) + \lambda \gamma \| \text{cov}(\hat{\mathbf{Y}}, \mathbf{A}) - \text{cov}(\hat{\mathbf{Y}}, \tilde{\mathbf{A}}) \|^2$$

$$+ \frac{\lambda}{|\mathcal{I}_{\text{train}}|} \sum_{i \in \mathcal{I}_{\text{train}}} -\left[ \log\left( \hat{d}_{\theta_d}\left( \hat{f}_{\theta_f}(X_i), \tilde{A}_i, Y_i \right) \right) + \log\left( 1 - \hat{d}_{\theta_d}\left( \hat{f}_{\theta_f}(X_i), A_i, Y_i \right) \right) \right]$$

$$\theta_f \leftarrow \theta_f - \mu \nabla_{\theta_f} \mathcal{J}_f(\theta_f)$$

**Output**: Predictive model $\hat{f}_{\theta_f}(\cdot)$ approximately satisfying equalized odds.

---

conditional density of $A \mid Y$. For example, when the sensitive attribute of interest is binary, we apply Bayes' rule and obtain

$$\mathbb{P}\{A = 1 | Y = y\} = \frac{\mathbb{P}\{Y = y \mid A = 1\}\mathbb{P}\{A = 1\}}{\mathbb{P}\{Y = y \mid A = 1\}\mathbb{P}\{A = 1\} + \mathbb{P}\{Y = y \mid A = 0\}\mathbb{P}\{A = 0\}}. \quad (4)$$

All the terms in the above equation are straightforward to estimate; in practice, we approximate terms of the form $\mathbb{P}\{Y = y \mid A = a\}$ using a linear kernel density estimation. For a non-binary sensitive attribute $A$, the fair dummies $\tilde{A}$ can be sampled by estimating the conditional distribution of $A \mid Y$. For instance, one can use quantile regression for a continuous variable $A$.

## 3 Validating equalized odds

Once we have a fixed predictive model $\hat{f}(\cdot)$ in hand (for example, a model fit on a separate training set), it is important to carefully evaluate whether equalized odds is violated on test points $\{(X_i, A_i, Y_i)\}_{i \in \mathcal{I}_{\text{test}}}$. To this end, we develop a hypothesis test for the relation (1). Our test leverages once again the fair dummies $\tilde{A}_i$, but we emphasize that it applies to any prediction rule, not just those trained with our proposed fitting method. The idea is straightforward: we generate many instances of the test fair dummies $\tilde{\mathbf{A}}$ and compare the observed test data $(\hat{\mathbf{Y}}, \mathbf{A}, \mathbf{Y})$ to those with the dummy attributes $(\hat{\mathbf{Y}}, \tilde{\mathbf{A}}, \mathbf{Y})$, since the latter triple obeys equalized odds. One can compare these distributions with any test statistic to obtain a valid hypothesis test; this is a special case of the conditional randomization test of [20]. In Algorithm 2 below, we present a version of this general test using [21] to form test statistic based on a deep neural network $\hat{r}(\cdot)$. Invoking [20], the output of the test is a p-value for the hypothesis that equalized odds holds:

**Proposition 2.** *Suppose the test observations $(Y_i, X_i, A_i)$ for $i \in \mathcal{I}_{\text{test}}$ are i.i.d.. Set $\hat{Y}_i = \hat{f}(X_i)$ for a fixed function $\hat{f}(\cdot)$ and construct independently distributed fair dummies $\tilde{A}_i$ as in Proposition 1. If equalized odds holds for each $i$, i.e., $\hat{Y}_i \perp\!\!\!\perp A_i \mid Y_i$, then the distribution of the output $p_v$ of Algorithm 2 stochastically dominates the uniform distribution; in other words, it is a valid p-value.*

We reiterate that this holds for any choice of the test statistic $T(\cdot)$, so we next discuss a good all-around choice. For problems with a continuous response $Y \in \mathbb{R}$ and prediction $\hat{Y} \in \mathbb{R}$, we define the

**Algorithm 2** The Fair Dummies Test

**Input**: Data $\{(\hat{Y}_i, A_i, Y_i)\}, i \in \mathcal{I}_{\text{test}}$

1: Split $\mathcal{I}_{\text{test}}$ into disjoint subsets $\mathcal{I}_1$ and $\mathcal{I}_2$.
2: Fit a model $\hat{r}(A_i, Y_i)$ on $\{(\hat{Y}_i, A_i, Y_i) : i \in \mathcal{I}_1\}$, aiming to predict $\hat{Y}_i$ given $(A_i, Y_i)$.
3: Compute the test statistic on the validation set: $t^* = \frac{1}{|\mathcal{I}_2|} \sum_{i \in \mathcal{I}_2} T(\hat{Y}_i, Y_i, \hat{r}(A_i, Y_i))$.
4: **for** $k = 1, \ldots, K$ **do**
5:    Sample a fresh copy of the fair dummies $\tilde{A}_i \sim P_{A|Y}(A_i \mid Y_i),\ i \in \mathcal{I}_2$.
6:    Compute the test statistic using the fair dummies: $t^{(k)} = \frac{1}{|\mathcal{I}_2|} \sum_{i \in \mathcal{I}_2} T(\hat{Y}_i, Y_i, \hat{r}(\tilde{A}_i, Y_i))$.
7: Compute the quantile of the true statistic $t^*$ among the fair dummy statistics $t_1, \ldots, t_K$:

$$p_v = \frac{1 + \#\{k : t^* \geq t^{(k)}\}}{K + 1}.$$

**Output**: A p-value $p_v$ for the hypothesis that (1) holds, valid under the assumptions of Proposition 2.

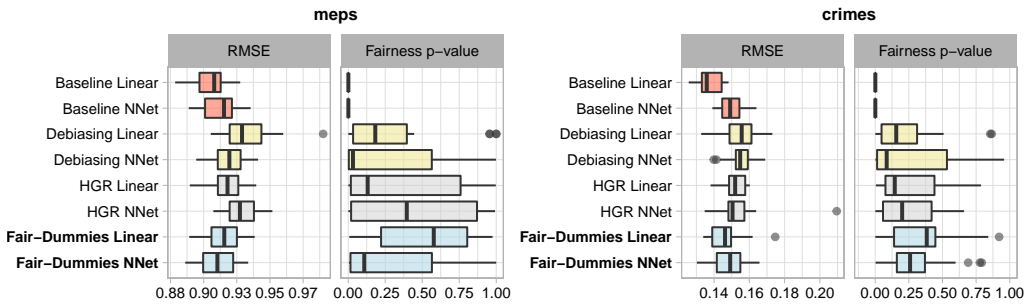

Figure 2: Real data regression experiments on the MEPS (left) and Communities and Crimes (right) data sets. The results are shown over 20 random splits of the data. Each figure presents the RMSE as well as the equalized odds p-values obtained with the fair dummies test.

test statistic as the squared error function, $T(\hat{Y}_i, Y_i, \hat{r}(A_i, Y_i)) = (\hat{Y}_i - \hat{r}(A_i, Y_i))^2$. Here, $\hat{r}(\cdot)$ can be any model predicting $\hat{Y}_i \in \mathbb{R}$ from $(A_i, Y_i)$; we use a two-layer neural network in our experiments. We describe a similar test statistic for multi-class classification in Supplementary Section S2.

As a final remark, note that a naive resampling scheme where we instead randomly resample $A$ unconditionally would result in a test of the hypothesis $\hat{Y} \perp\!\!\!\perp A$, a property called *demographic parity* [2, 3]. The fair dummies test, in contrast, is able to test a richer notion of fairness by resampling in a way that reflects the structure of the more sophisticated equalized odds property.

## 4 Experiments

We now evaluate our proposed fitting method in real data experiments. We compare our approach to two recently published methods, adversarial debiasing [9] and HGR [10], demonstrating moderately improved performance. While our fitting algorithm also applies to binary classification, we only consider regression and multi-class classification tasks here because there are very few available techniques for such problems. In all experiments, we randomly split the data into a training set (60%), a hold-out set (20%) to fit the test statistic for the fair-dummies test, and a test set (20%) to evaluate their performance. We do not use the sensitive attribute as a feature in our experiments. See Supplementary Section S3 for a synthetic experiment in which we include the sensitive attribute as an additional feature and further discussion of this point. The software is available online at `https://github.com/yromano/fair_dummies`.

### 4.1 Real data: regression

We begin with experiments on two data sets with real-valued responses: the 2016 Medical Expenditure Panel Survey (MEPS), where we seek to predict medical usage based on demographic variables,

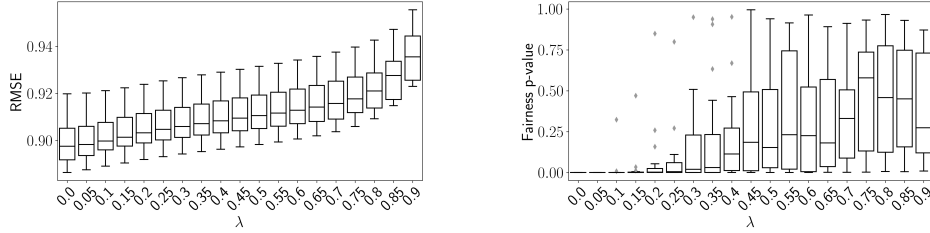

Figure 3: The effect of the regularization parameter on model training. The results are reported across 20 random splits of the MEPS data. Left: accuracy. Right: fair dummies test p-value.

and the widely used UCI Communities and Crime data set, where we seek to predict violent crime levels from census and police data. See Supplementary Section S5.1 for more details. Decision makers may wish to predict medical usage or crime rates to better allocate medical funding, social programs, police resources and so on [e.g., 22], but such information must be treated carefully. For both data sets we use race information as a binary sensitive attribute, and it is not used as a covariate for the predictive model. An equalized odds model in this context can add a layer of protection against possible misuse of the model predictions by downstream agents: any two people (neighborhoods) with the same underlying medical usage (crime rate) would be treated the same by the model, regardless of racial makeup. Further care is still required to ensure that such a model is deployed ethically, but equalized odds serves as a useful safeguard.

We will consider two base predictors: a linear model and a neural network. As fairness-unaware baselines, we fit each of the above by minimizing the MSE, without any fairness promoting penalty. We also use each of the base regression models together with the *adversarial debiasing* method [9], the *HGR* method [10], and our proposed method; see Supplementary Section S6 for technical details. The methods that promote equalized odds, including our own, each have many hyperparameters, and we find it challenging to automate the task of finding a set of parameters that maximizes accuracy while approximately achieving equalized odds, as also observed in [9]. Therefore, we choose to tune the set of parameters of each method only once and treat the chosen set as fixed in future experiments; see Supplementary Section S6.1 for a full description of the tuning of each method.

The performance of these methods is summarized in Figure 2. We observe that the p-values of the two fairness-unaware baseline algorithms are small, indicating that the underlying predictions may not satisfy the equalized odds requirement. In contrast, adversarial debiasing, HGR, and our approach are all better aligned with the equalized odds criterion as the p-values of the fair dummies test are dispersed on the $[0, 1]$ range. Turning to the predictive accuracy, we find that the fairness-aware methods perform similarly to each other, although our proposed methods perform a little better than the alternatives. Each of the fairness-aware models have slightly worse RMSE than the corresponding fairness-unaware baselines, as expected.

Figure 3 presents the trade-off between accuracy and fairness (measured by our p-value for the equalized odds requirement) across values of the regularization parameter $\lambda$ for the MEPS data set. We fit a neural network model with the same hyperparameters as in Figure 2. The left panel of Figure 3 shows that an increase in $\lambda$ reduces the RMSE of the trained model, as expected. The right panel demonstrates that the fair dummies test p-values increase with more regularization and that for small values of $\lambda$, the proposed test detects violations of equalized odds.

## 4.2   Real data: multi-class classification

Next, we consider a multi-class classification example using the UCI Nursery data set, where we aim to rank nursery school applications based on family information. The response has four classes and we use financial standing as a binary sensitive attribute. See Supplementary Section S5.2 for more details. Similar to our regression experiments, we use a linear multi-class logistic regression and neural network as fairness-unaware baseline algorithms. As before, we also fit predictive models using our proposed method and compare the results to those from adversarial debiasing and HGR. The latter only handles one-dimensional $\hat{Y}$, so we adapted it to the multi-class setting by evaluating the penalty separately on each element of the vector of class-probabilities $\hat{Y} \in \mathbb{R}^L$ and summing all $L$ of the penalty scores. See Supplementary Section S6 for additional details.

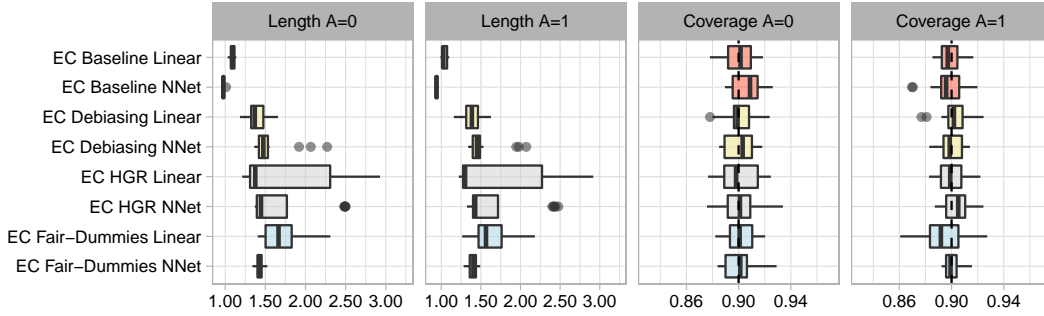

Figure 5: Classification experiment on the Nursery data set. The results are shown for 20 random splits. Left to right: average size of prediction set per group, coverage per group (target 90%).

We report the results in Figure 4. The p-values that correspond to the fairness-unaware baseline algorithms are close to zero, indicating that these methods violate the equalized odds requirement. In contrast, HGR, adversarial debiasing, and our method lead to a nice spread of the p-values over the $[0, 1]$ range, with the exception of adversarial debiasing with the linear model which appears to violate equalized odds. Turning to the prediction error, when forcing the equalized odds criterion the statistical efficiency is significantly reduced compared to the fairness-unaware baselines, and since the linear adversarial debiasing method violates the equalized odds property, our method has the best performance among procedures that seem to satisfy this criterion.

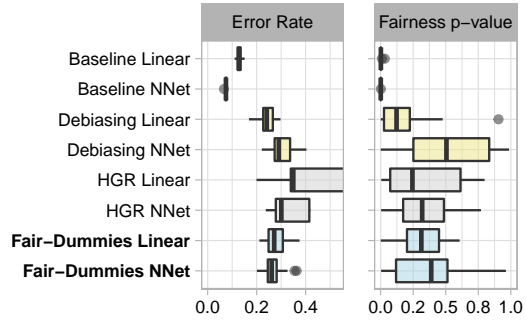

Figure 4: Classification experiment on the Nursery data set. Results are shown for 20 random splits. Left: misclassification error. Right: fair dummies test p-value. Large values for HGR not shown.

## 5 Evaluating performance with uncertainty sets

Quantifying uncertainty in predictive modeling is essential, and, as a final case study, we revisit the previous data set with a new metric based on prediction sets. In particular, using the *equalized coverage* method [23], we create predictive sets $C(X, A) \subseteq \{1, 2, \ldots, L\}$ that are guaranteed to contain the unknown response $Y$ with probability 90%. To ensure the prediction sets are unbiased to the sensitive attribute, the coverage property is made to hold identically across values of $A = a$:

$$\mathbb{P}\{Y \in C(X, A) \mid A = a\} \geq 90\% \qquad \text{for all } a \in \{0, 1\}.$$

Such sets can be created using any base predictor, and we report on these sets for the methods previously discussed in Figure 5; see Supplementary Section S7. We observe that all methods obtain exactly 90% coverage per group, as guaranteed by the theory [23]. To compare the statistical efficiency, we look at the size of the prediction sets; smaller size corresponds to more precise predictions. Among the prediction rules that approximately satisfy equalized odds, a neural network trained with our proposed penalty performs the best (recall from Figure 4 that the linear method with adversarial debiasing violates equalized odds in this case).

## 6 Discussion

### 6.1 Connection with other notions of fairness

Up to now, our work has dealt with equalized odds, and we pause here to discuss its place within the broader landscape of algorithmic fairness. There are four groups of fairness formalisms in the literature. First, *fairness through unawareness* asks that you fit the model without the sensitive

attribute $A$. This notion fails in that other features in use by the model may serve as a proxy for $A$. Note that if you simply flipped $A$ randomly and then fit a model, you would end up with a model obeying fairness through unawareness. In contrast, our proposed fitting algorithm uses a more nuanced resampling scheme to target a sharper notion of fairness, equalized odds. Second, *individual fairness* approaches require that similar individuals (according to some chosen metric) get similar predictions [1]. Third, *statistical parity* notions require that some conditional independence relation is satisfied in the population. Equalized odds is one example, and two others are *calibration* and *demographic parity*—see [2, 3, 8] for a rich discussion of these three. Lastly, there are a set of *causal fairness* notions that are gaining recognition. *Counterfactual fairness* requires that the predictions will remain unchanged after an intervention is carried out on the sensitive attribute and the resulting changes are propagated through the causal graph [6, 24]. Related notions define fairness as the blocking of causal paths deemed to be unfair by the researcher [7, 25, 26]. The relationship between the causal notions and statistical parity notions is discussed in detail in [24].

While we focus on equalized odds for concreteness, our proposal is at its core about training models to satisfy a conditional independence relation among the predictions, response, and sensitive attribute(s). Our method can be easily adapted to promote the calibration property in training. Some causal notions of fairness can also be cast as more complex conditional independence relations. Extending our work as a hypothesis test and model fitting algorithm for counterfactual fairness, say, is possible, but is more complex because the dimensionality would be larger than in our setting. Nonetheless, such an extension appears within reach, and we view it as a promising next step.

## 6.2 Looking forward

In this work we presented a novel method for fitting models that approximately satisfy the equalized odds criterion, as well as a rigorous statistical test to detect violations of this property. The latter is the first of its kind, and we view it as an important step toward understanding the equalized odds property with complex models. Returning to the former, a handful of other approaches have been proposed, and we demonstrated similar or better performance to state-the-art methods in our numerical experiments. Beyond statistical efficiency, we wish to highlight the flexibility of our proposed approach. Our penalization scheme can be used with any discriminator or two sample test, any loss function, any architecture, any training algorithm, and so on, with minimal modification. Moreover, the inclusion of the second moment penalty makes our scheme stable, alleviating the sensitivity to the choice of hyperparameters. From a mathematical perspective, the synthetic data allows us to translate the problem of promoting and testing a conditional independence relation to the potentially more tractable problem of promoting and testing equality in distribution of two samples. We expect this reframing will be useful broadly within algorithmic fairness. Lastly, we point out our procedure applies more generally to the task of fitting a predictive model while promoting a conditional independence relation [e.g., 13], and leveraging this same technique in domains other than algorithmic fairness is a promising direction for future work.

We view our proposal as a way to *move beyond mean squared error*; with modern flexible methods, there are often many prediction rules that achieve indistinguishable predictive performance, but they may have different properties with respect to robustness, fairness, and so on. When there is a rich enough set of good prediction rules, we can choose one that approximately satisfies the equalized odds property. Nonetheless, we point out two potential problems with exclusively focusing on the equalized odds criterion. First, it is well-known that forcing a learned model to satisfy the equalized odds can lead to decreased predictive performance [27, 28, 2, 3, 29, 30], as illustrated in Figure 3. Demanding that the equalized odds is exactly satisfied may force us to intentionally destroy information, as clearly seen in the algorithms for binary prediction rules in [8, 11, 12, 9, 10], and as implicitly happens in some of our experiments. Second, for regression and multi-class classification problems, there is no known way to certify that a prediction rule exactly satisfies equalized odds or to precisely bound the violation from this ideal, so the resulting prediction rules do not come with any formal guarantee. Both of these issues are alleviated when we return uncertainty intervals that satisfy the equalized coverage property, as shown in Section 5. With this approach, we regularize models towards equalized odds to the extent desired, while returning uncertainty sets valid for each group separately to accurately convey any difference in performance across the groups. Importantly, this gives an interpretable, finite-sample fairness guarantee only relying on the assumption of i.i.d. data. For these reasons, we see the combination of an (approximately) equalized odds model with equalized coverage predictive sets as an attractive combination for models in high-stakes deployments.

## Broader Impact

This work aims to build tools for fair, reliable machine learning algorithms for high-stakes decisions—an essential task for the ethical use of machine learning. The immediate positive outcome from this work is a new algorithm for training algorithms to satisfy the equalized odds property. Our technique explicitly detects biases in a learned model to alert the analyst to any imbalanced performance, while training a model to seek equal performance, when possible. One technical point of failure is that our hypothesis test assumes i.i.d. data. If this assumption fails, the test may lead to incorrect and potentially biased results. Furthermore, while our hypothesis test can detect some violations of the equalized odds property, it is not guaranteed to detect any such violation. Lastly, one possible negative impact is that with the increasing availability of so-called "fair" training algorithms, researchers will accept that an algorithm is fair or ethical without sufficient scrutiny. We emphasize that ethical machine learning must be viewed as an important unsolved problem that requires both further algorithmic and conceptual advances, as well as rigorous, critical thought on the part of the researcher in each new setting.

## Acknowledgments and Disclosure of Funding

E. C. was partially supported by the Office of Naval Research grant N00014-20-12157, and by the National Science Foundation grants DMS 1712800 and OAC 1934578. He thanks Rina Barber and Chiara Sabatti for useful discussions related to this project. S. B. was supported by NSF under grant DMS 1712800 and a Ric Weiland Graduate Fellowship. Y. R. was supported by the Army Research Office (ARO) under grant W911NF-17-1-0304. Y. R. thanks the Zuckerman Institute, ISEF Foundation, the Viterbi Fellowship, Technion, and the Koret Foundation, for providing additional research support.

## Footnotes

[1]This means that we can write $\tilde{A} = h(Y, \epsilon)$ for some function $h(\cdot)$, where the random variable $\epsilon$ is independent of everything else.

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
