[Supplementary Material]

# Achieving Equalized Odds by Resampling Sensitive Attributes

# Supplementary Material

**Yaniv Romano**
Department of Statistics
Stanford University
Stanford, CA, USA
yromano@stanford.edu

**Stephen Bates**
Department of Statistics
Stanford University
Stanford, CA, USA
stephenbates@stanford.edu

**Emmanuel J. Candès**
Departments of Mathematics
and of Statistics
Stanford University
Stanford, CA, USA
candes@stanford.edu

## S1 Proofs

*Proof of Proposition 1.* The "if" direction is immediate. For the reverse direction, taking discrete random variables for simplicity, we have

$$
\begin{aligned}
\mathbb{P}(\hat{Y} = \hat{y}, A = a, Y = y) &= \mathbb{P}(\hat{Y} = \hat{y}, A = a \mid Y = y) \cdot \mathbb{P}(Y = y) \\
&= \mathbb{P}(\hat{Y} = \hat{y} \mid Y = y) \cdot \mathbb{P}(A = a \mid Y = y) \cdot \mathbb{P}(Y = y) \\
&= \mathbb{P}(\hat{Y} = \hat{y} \mid Y = y) \cdot \mathbb{P}(\tilde{A} = a \mid Y = y) \cdot \mathbb{P}(Y = y) \\
&= \mathbb{P}(\hat{Y} = \hat{y}, \tilde{A} = a \mid Y = y) \cdot \mathbb{P}(Y = y) \\
&= \mathbb{P}(\hat{Y} = \hat{y}, \tilde{A} = a, Y = y)
\end{aligned}
$$

$\square$

*Proof of Proposition 2.* The proposed test is an instance of the Holdout Randomization Test [1], which is in turn a special case of the Conditional Randomization Test [2], so the result follows directly from Lemma 4.1 of [2]. $\square$

## S2 Test statistics for multi-class classification

In this section, we give the details of the fair dummies test (Algorithm 2) for multi-class classification. Here, with response $Y \in \{1, \ldots, L\}$ and class probability estimates $\hat{Y} \in \mathbb{R}^L$, let $\hat{Y}^Y \in \mathbb{R}$ be the variable located in the $Y^{\text{th}}$ entry of $\hat{Y}$. Similar to the regression case, we fit a predictive model $\hat{r}(A_i, Y_i) \in \mathbb{R}$, aiming to predict the estimated class probability $\hat{Y}_i^{Y_i}$ given the pair $(A_i, Y_i)$ by minimizing the cross entropy loss function. (We use a one-hot encoding for $Y_i$.) This function is then used to formulate our final test statistic:

$$
T(\hat{Y}_i, Y_i, \hat{r}(A_i, Y_i)) = -\hat{Y}_i^{Y_i} \log(\hat{r}(A_i, Y_i)) - (1 - \hat{Y}_i^{Y_i}) \log(1 - \hat{r}(A_i, Y_i)).
$$

Another reasonable statistic for this setting would be to use the whole vector of class probabilities together with the multi-class cross-entropy loss, but we found that the above is more powerful at detecting violations of equalized odds.

(a) Baseline, majority.　　(b) Baseline, minority.　　(c) Proposed, majority.　　(d) Proposed, minority.

Figure S1: Predictions obtained from models that are applied on a feature vector that *does not include* the sensitive attribute $A$. (a,b) fairness-unaware baseline neural network model; (c,d) equalized-odds-aware neural network model.

(a) Baseline, majority.　　(b) Baseline, minority.　　(c) Proposed, majority.　　(d) Proposed, minority.

Figure S2: Predictions obtained from models that are applied on a feature vector that *does include* the sensitive attribute $A$; other details as in Figure S1.

## S3　Synthetic experiment: using the sensitive attribute as a feature

Our approach also applies in situations where the feature vector contains the sensitive attribute. To illustrate this point, we simulate data with two features $X_1, X_2 \sim \mathcal{N}(0,1)$ and two groups $A \in \{0,1\}$. We create an unbalanced population, where 90% of the samples are from the majority group $A = 0$. The response variables follow a linear model, $Y = X^\top \beta_A + \epsilon$, where $\beta_0 = (2,1)$, $\beta_1 = (1,2)$, and $\epsilon \sim \mathcal{N}(0,1)$ is a noise term. Similarly to Section 1.1, the conditional distribution is the same for the majority and minority groups (again, up to a permutation of the coordinates), and an ideal model should perform similarly on the two groups. In contrast with Section 1.1, here, the two groups are indistinguishable if we only have access to the pair $(X_1, X_2)$. Therefore, a predictive rule that has access to $A$ in addition to $(X_1, X_2)$ should attain better performance compared to a model that is blind to the sensitive attribute.

We generate 500 training points and fit two baseline neural network models. The first aims to predict $Y$ given $(X_1, X_2)$, while the second also makes use of $A$ to improve accuracy. We train the two baseline models by minimizing the mean squared error. To evaluate their performance, we generate an independent test set of 2000 samples, which we use to compute the RMSE as well as to evaluate the fairness p-value using Algorithm 2. To this end, we generate additional independent set of 2000 points to fit the function $\hat{r}(\cdot)$ from Algorithm 2. We summarize the results in Table S1 and visualize the predictions obtained by the two baseline models in Figures S1 and S2. As can be seen, the two models perform better on the majority group $A = 0$, where the one that includes $A$ as a feature achieves the best RMSE. Following that table, the two baseline methods violate the equalized odds requirement—the fairness p-values in both cases are small and equal to 0.001.

We next illustrate the effect of our penalty function by fitting two fairness-aware models, with and without including $A$ as a feature. The new models have the same architecture as the baseline methods, and both are trained and tested using the same data sets used to fit and evaluate the baseline models. In Table S1 we see that the fairness-aware models pass the fair dummies test and that the proposed models result in more balanced performance across the two groups; see also Figures S1 and S2. Here, the best model is the one whose feature vector contains the sensitive attribute $A$ in addition to the pair $(X_1, X_2)$, as expected.

| | RMSE | | | Fairness p-value |
|---|---|---|---|---|
| | Minority group A=0 | Majority group A=1 | Overall | |
| Baseline without $A$ | 1.020 | 1.659 | 1.098 | 0.001 |
| Proposed without $A$ | 1.303 | 1.240 | 1.297 | 0.629 |
| Baseline with $A$ | 1.007 | 1.175 | 1.025 | 0.001 |
| Proposed with $A$ | 1.061 | 1.053 | 1.061 | 0.402 |

Table S1: Performance of baseline and our equalized-odds-aware model fitting on simulated data. 'Baseline without $A$' and 'Proposed without $A$' do not use the sensitive attribute $A$ in the model whereas 'Baseline with $A$' and 'Proposed with $A$' do.

## S4 Other approaches to testing equalized odds

The reader may wonder: is the fair dummies test the only way to test equalized odds? We give a brief commentary here on other possible approaches. First, if $A$, $\hat{Y}$, and $Y$ are all categorical, there are generic approaches for testing any conditional independence relation [3, 4]. Although this approach would in principle apply, it requires that the number of categories is very small to have power for realistic sample sizes. Moreover, this approach would not generalized to regression problems. For regression problems, one can instead imagine using a test based on a linear model for $\hat{Y}$ given $Y$ and $A$ (or a log-transformed version thereof) where one checks if an interaction term of $Y$ with $A$ is zero. The validity of this approach requires the assumption that $\hat{Y}$ follows a linear model given on $Y$; otherwise, it is not known to control the type-I error at level $\alpha$. In contrast, the fair dummies test requires no such linearity assumption.

## S5 Data sets

### S5.1 Regression

For regression problems, we compare the performance of our methods to adversarial debiasing [5] and HGR [6] on the following two data sets:

- The 2016 Medical Expenditure Panel Survey (MEPS).[1] Here, the goal is to predict the utilization of medical services based on features such as the individual's age, marital status, race, poverty status, and functional limitations. After pre-processing the data as in [7], there are 15656 samples and 138 features. We take race as the binary sensitive attribute—there are 9640 white individuals and 6016 non-white individuals. Note that MEPS data is subject to usage rules. We downloaded the data set using conformalized quantile regression [7] software package, available online.[2]

- Communities and Crime data set.[3] The goal is to estimate the number of violent crimes for U.S. cities given the median family income, per capita number of police officers, percent of officers assigned to drug units, and so on. We clean the data according to [6], resulting in 1994 observations of 121 variables. Race information is again used as the as sensitive attribute, with 784 observations from communities whose percentage of African American is above 10% and 1210 observations from other communities.

### S5.2 Multi-class classification

The Nursery data contains information on nursery school applicants.[4] The task is to rank applications based on features such as the parents' occupation, family structure, and financial standing. The original data set contains five classes, however, after cleaning and rearranging the data we remain with

four classes: children who are (1) "not recommended", (2) "very recommended", (3) "prioritized", and (4) "specifically prioritized" to join the nursery. In total, the data set contains 12958 examples and 13 features. We use the financial status as a sensitive attribute; applicants with "inconvenient" standing are assigned to group $A = 0$ (6478 samples) and those with "convenient" status are assigned to group $A = 1$ (6480 samples).

## S6    Further information about the learning algorithms

### S6.1    Hyper-parameter tuning

To successfully deploy the learning algorithms presented in Section 4, we must tune various hyperparameters, such as the equalized odds penalty weight, learning rate, batch size, and number of epochs. This task is particularly challenging because we have a multi-criteria objective: the goal is not only to maximize accuracy but also to pass the fair dummies test, i.e. approximately achieve equalized odds. In our experiments, we find the best set of parameters using 10 fold cross validation, optimizing the accuracy-fairness objective. Since this process is computationally expensive and partly manual, in practice, we tune the hyperparameters only once using cross validation on the entire data set and then treat the chosen set as fixed for the rest of the experiments. The drawback of this approach is that it may suffer from over-fitting, since we test on the same data used to tune the hyperparameters. To mitigate this problem, in Section 4, we compare the performance metrics of the different algorithms on data splits that are different than the ones used to tune the parameters; some optimism, however, remains. In any case, this same tuning scheme is used for all methods, ensuring that the comparisons are meaningful.

### S6.2    Implementation details

The source code implementing the experiments is available for download online at `https://github.com/yromano/fair_dummies`. We run the experiments on our local cluster.

**Regression**

Our regression experiments build on two base learning algorithms, which are then combined with HGR, adversarial debiasing, and our framework to yield eight methods:

- Baseline Linear: we fit a linear model by minimizing the MSE loss function, using the stochastic gradient descent optimizer with a learning rate and number of epochs in $\{0.01, 0.1\}$ and $\{100, 200, 400, 600, 1000, 2000, 3000, 4000\}$, respectively. We normalize the features to have zero mean and unit variance using the training data.

- Baseline NNet: we fit a two layer neural network with a 64-dimensional hidden layer and ReLU nonlinearity function. The network is optimized by minimizing the MSE, following the same fitting strategy described in Baseline Linear.

- Debiasing Linear and Debiasing NNet: the predictors are formulated as described in the baseline algorithms. Here, we follow the implementation provided in `https://github.com/equialgo/fairness-in-ml` and design the adversary as a four-layer neural network with hidden layers of size 32 and ReLU nonlinearities. Since the sensitive attribute is binary, we apply the sigmoid function on the output of the last layer. We use the Adam optimizer [8] for training, with a learning rate in $\{0.001, 0.01, 0.1\}$ and a minibatch size in $\{64, 128\}$. We also follow the pre-training strategy suggested in [5] and fit separately the predictor and adversary for a number of epochs in $\{2, 4, 10, 20, 30, 40\}$. Then, the two pre-trained models are fitted interchangeably for additional $\{50, 100, 200, 300, 400\}$ epochs. The weight on the equalized odds penalty is selected from $\{0.2, 0.3, 0.4, 0.5, 0.6, 0.7, 0.8, 0.9, 0.95, 0.99\}$.

- HGR Linear and HGR NNet: we again use architectures identical to those of the baseline models. As suggested in [6], we use the Adam optimizer with a minibatch size in $\{128, 256\}$, learning rate in $\{0.001, 0.01\}$, and the number of epochs in $\{10, 15, 20, 30, 40, 50, 80, 100\}$. The HGR function is implemented in `https://github.com/criteo-research/continuous-fairness` and we select the weight penalty from the $\{0.1, 0.2, 0.3, 0.4, 0.5, 0.6, 0.7, 0.8, 0.9\}$ range.

- Fair-Dummies Linear and Fair-Dummies NNet: we fit predictors that have the same structure as the baseline algorithms, with our proposed regularization. The discriminator is implemented as a two-layer neural network with a hidden layer of size 30 and ReLU non-linearities. We use the stochastic gradient descent optimizer, with a fixed learning rate of 0.01. We use the same optimizer for the classifier, with the same learning rate, except for the addition of a momentum term with value 0.9. The number of epochs is chosen from the $\{20, 30, 40, 50, 80, 100\}$ range, and the number of gradient steps ($N_g$ in Algorithm 1) is selected from the range of $\{40, 50, 60, 70, 80\}$. The weight on the equalized odds penalty is selected from $\{0.4, 0.5, 0.6, 0.7, 0.8, 0.9\}$ ($\lambda$ in Algorithm 1), and the second moment term ($\gamma$ in Algorithm 1) is chosen from $\{1, 10, 20\}$.

The predictive model $\hat{r}(\cdot)$, defining the test statistics in the fair dummies test (see Section 3), is formulated as a two-layer neural network, with a hidden dimension of size 64, and dropout layer with rate $1/2$. We use stochastic gradient descent to fit the network, run for 200 epochs with a minibatch of size 128 and a fixed momentum term with weight 0.9.

**Multi-class classification**

Our experiments are again based on two underlying predictive models which are regularized using fairness-aware methodologies:

- Baseline Linear: we fit a linear model by minimizing the cross entropy loss function. We use the Adam optimizer, with a minibatch size of 32. We choose the learning rate, and number of epochs from the range of $\{0.001, 0.01, 0.1\}$, and $\{20, 40, 60, 80, 100\}$, respectively. We normalize the features to have zero mean and unit variance using the training data.

- Baseline NNet: we fit a two layer neural network with a 64-dimensional hidden layer, ReLU nonlinearity function, and dropout regularization with rate $1/2$. We use the same optimization strategy as above above.

- Debiasing Linear and Debiasing NNet: we form classifiers as in the baseline algorithms. Similarly to the regression setting, we rely on the implementation from `https://github.com/equialgo/fairness-in-ml`. We use the same adversary as described in the regression setting. Training is done via the Adam optimizer, with a fixed learning rate that is equal to 0.5 and minibatches of size 32. We again apply the pre-training strategy [5] and fit separately the predictor and adversary for number of epochs from the range of $\{1, 2\}$. The adversarial training is then repeated for $\{20, 40, 60, 100, 200\}$ epochs. The weight on the equalized odds penalty is selected from $\{0.9, 0.99, 0.999, 0.9999, 0.99999, 0.999999\}$.

- HGR Linear and HGR NNet: we again take classifiers as in the baseline models. To fit them, we apply the Adam optimizer with a mini-batch size of 128, learning rate in the range of $\{0.001, 0.01\}$, number of epochs selected from $\{10, 20, 30, 40, 50\}$. The HGR penalty weight is selected in the range of $\{0.9, 0.91, 0.92, 0.93, 0.94, 0.95, 0.96, 0.97, 0.98, 0.99\}$.

- Fair-Dummies Linear and Fair-Dummies NNet: we again take classifiers with the same structure as the baseline algorithms. The adversary is implemented as a four-layer neural network with a 32-dimensional hidden layer and ReLU nonlinearity. We use the Adam optimizer, with a fixed learning rate that is equal to 0.5. The number of epochs is fixed and equal to 50. The number of gradient steps $N_g$ is selected in the range of $\{1, 2\}$, and the weight on the equalized odds penalty is selected from $\{0.9, 0.99, 0.999, 0.9999, 0.99999, 0.999999\}$ for $\lambda$ and from $\{0.01, 0.001, 0001, 0.00001\}$ for $\gamma$.

The fair dummies test statistics is again evaluated using a predictive model $\hat{r}(\cdot)$ that is implemented as a neural network. We use the same architecture and learning strategy as in the regression setup, with the addition of a sigmoid function as the last layer.

## S7    Further details on equalized coverage

We now turn to a few details of the equalized coverage prediction sets from Section 5. First, note that the prediction sets here are the same as frequentist confidence intervals, except that we are trying to cover the observed value on a fresh data point, rather than a model parameter (the usual target in frequentist inference). See [9] for more discussion of this point. Turning to our

experiments, we use the software package provided by [10], which is available online at `https://github.com/yromano/cqr`. While equalized coverage [10] is presented for regression problems, it is straightforward to extend this method to multi-class classification tasks. To this end, we follow split conformal prediction [11] and randomly split the data into a proper training set (60%), a hold-out calibration set (20%), and a test set (20%). We use the same predictive models from Section 4.2, which are fitted to the whole proper training data, providing estimates for class probabilities. The examples $\{(X_i, A_i, Y_i)\}$ that belong to the calibration set are then used to construct the prediction sets for the test points. Specifically, following the notations from Section 3, we deploy the popular inverse probability conformity score [12], given by $1 - \hat{Y}_i^{Y_i}$. Here, $\hat{Y}_i = \hat{f}(X_i) \in \mathbb{R}^L$ and the variable $\hat{Y}_i^{Y_i} \in \mathbb{R}$ is the estimated probability that the calibration example $X_i$ belongs to class $Y_i$.

## Footnotes

[1] https://meps.ahrq.gov/mepsweb/data_stats/download_data_files_detail.jsp?cboPufNumber=HC-192

[2] https://github.com/yromano/cqr

[3] http://archive.ics.uci.edu/ml/datasets/communities+and+crime

[4] https://archive.ics.uci.edu/ml/datasets/nursery