[Reviews · NeurIPS 2020]

Review 1

Summary and Contributions: This paper presents an approach for better satisfying equalized odds fairness requirements by resampling sensitive attributes to satisfy equalized odds and then using the divergence from this distribution in the training optimization. The paper also presents a statistical test for equalized odds based on similar sensitive attribute resampling. It then show experimentally that the proposed approach achieves lower error and better performance on the statistical test for equalized odds.

Strengths: The approach is quite general; the paper addresses regression and multiclass classification, which demonstrate this generality. The contribution of a general statistical test for equalized odds is also valuable. The topic of fairness for machine learning is very important in the ML community as methods are being increasingly applied to social prediction/decision-making tasks.

Weaknesses: While only assessing p-values may match realistic deployments, since the full ground truth is available for the experiments, it would be useful to also characterize the actual equalized odds violations. Since many hyperparameters typically exist for all methods, providing results showing the trade-offs between predictive loss and equalized odds violations using varying hyperparameter values would be beneficial.

Correctness: The paper appears to be correct.

Clarity: The paper is well written.

Relation to Prior Work: The paper compares against state-of-the-art methods for equalized odds in regression and multi-class prediction.

Reproducibility: Yes

Additional Feedback:


Review 2

Summary and Contributions: This paper presents a framework for learning predictive models that approximately satisfy the equalized odds notion of fairness both in regression and multi-class classification problems. They introduce a discrepancy functional that measures the violation of equalized odds. They also develop a hypothesis test to detect whether a prediction rule violates equalized odds. Empirically, their approach can achieve a similar fairness guarantee as baselines while the accuracy of their approach is higher.

Strengths: This paper investigates a classic fairness notion: equalized odds, and consider the real-valued or multi-class setting. The idea of resampling sensitive attributes is interesting.

Weaknesses: The proposed framework is heuristic and does not have a theoretical guarantee. The hypothesis test seems to just follow from the Holdout Randomization Test [1]. Also, I am not sure whether their proposed hypothesis test is better than the existing notion of equalized odds, which needs to be explained more clearly. Thank you for the response. I have no problem for the first two points. For the third point, I may not explain my question clearly before. My question is that in the literature, we can simply measure the violation of equalized odds by additive or multiplicative violations among groups. What are the advantages of hypothesis test compared to these simple ways? Also, I have mentioned the missing of comparison with causality literatures. I wish to see the response of this point. Then I can raise my score.

Correctness: The claims, method, and empirical methodology are correct.

Clarity: The paper is well written. From my view, I wish the authors state their novelty in the abstract more clearly. What is the main contribution on the model - the extension to the real-valued or multi-class response?

Relation to Prior Work: The comparison between their hypothesis test and the existing notion of equalized odds should be clear. Their idea seems to relate to the notion of causality, which has been investigated a lot in fair literature recently. A discussion with existing causality work is needed as I think.

Reproducibility: Yes

Additional Feedback:


Review 3

Summary and Contributions: The paper addresses algorithmic fairness setting with continual and multi-class outcomes. The authors propose to randomise/flip the values of the sensitive feature A to achieve equalised odds. This is in contrast to other fairness methods that perform flipping/resampling of the target labels Y, e.g. Kamiran and Calders [22]. The authors proposed to draw pseudo sensitive features as samples from p(A|Y) and use them instead of real sensitive features.

Strengths: Clarity of presentation is satisfactory. Experiments are presented and the baselines are appropriate. Randomising sensitive features is an intuitive idea that has been studied before. The intuition is that the target predictions should be independent given a random protected characteristic value. However, there are concerns with this approach discussed below.

Weaknesses: Intuitively randomising sensitive feature should lead to fairer results, however, fairness though unawareness poses a risk of unfairness by proxy as there are ways of predicting protected characteristic features from other features [Ruggieri et all, 2010, Adler et al 2016]. Also a continuous analog of fairness through unawareness [Dwark et al 2012] has been proposed via counterfactual fairness [Matt J. Kusner, et al, Counterfactual fairness, 2017]. In the counterfactual fairness, one has to estimate a dependency structure over the features, i.e. a causal graph, in order to create a counterfactual example when changing/flipping observational sensitive feature. To properly evaluate the contribution of the proposed approach, it has to be compared —methodologically and empirically — not only to fairness through unawareness, but also to counterfactual fairness approaches. Another concern is that very little information is dedicate to the analysis how to estimate p(A|Y). The described strategy in (4) could work for binary sensitive features, but how to generalise it to the continual features is not described. I wonder whether the assumption of equalised odds (2) being achieved automatically (because we sample sensitive features conditioned on the true labels Y and not the predictions Y_hat) excludes/discourages a perfect predictor, when Y_hat = Y. Minor comments: There is a lot of discussion on the discrepancy measure estimation in Section 2.2. It is surprising the authors did not utilise MMD GAN [Li et al, 2017, Binkowski et al 2018] in the proposed approach. Seems like a missed opportunity. ---------------------- Thank you for providing a rebuttal. I would like to confirm to the authors that I am fully aware of the work [6] Moritz Hardt, Eric Price, and Nati Srebro: Equality of opportunity in supervised learning, NeurIPS 2016. In my opinion, the most critical points (in this review and the one provided by the R2) regarding the background literature on causal learning and the counterfactual work (building a causal model and intervening on the sensitive attribute) have not been addressed. Let me recap once again: "To properly evaluate the contribution of the proposed approach, it has to be compared —methodologically and empirically — to counterfactual fairness approaches". The authors did not respond in any constructive way to this criticism. Secondly, I thought that the authors have compared their results to fairness through unawareness methods, when we do not use sensitive feature for training and testing the models. But after reading the rebuttal, I left uncertain whether the baseline methods entitled as "fairness unaware baselines" use sensitive information during training or not. Given the proposed approach is trying to change the sensitive feature, this is a crucial baseline to compare to. Finally, it has been stated "Certainly, other forms of randomization have been studied before, but randomizing [sensitive feature] conditionally on the observed Y is the crucial idea necessary to promote equalized odds in model fitting." We should be more mindful of the negative implications this approach might have in the context *beyond equalised odds*. The proposed method attempts to modify the sensitive information during training (fair dummies), and does not take responsibility to explain the model/outcomes. In contrast, in the causal models, when intervening on the sensitive feature, we can create counterfactual explanations that are mindful about the features and aim to explain the decisions.

Correctness: Appear so.

Clarity: Satisfactory.

Relation to Prior Work: Some concerns are described in the comments above.

Reproducibility: Yes

Additional Feedback:


Review 4

Summary and Contributions: The authors present an approach to developing fair predictive models. They adapt a number of works from the conditional independence testing and regularization literature to the equalized odds setting. They introduce three main techniques: (i) a regularizer, (ii) a hypothesis testing procedure, (iii) a confidence interval method. They show in experiments that the model performs favorably (in a fairness sense) to other models from the literature.

Strengths: The paper is well-written. It uses principled methods and thinks mostly rigorously about the correct way to approach equalized odds. The solutions proposed are practical, flexible, and general.

Weaknesses: The novelty here may be a little low. I am not personally familiar with the fairness literature, but I am very familiar with conditional independence testing methods. Given that, if someone had told me that the objective was to achieve equation (1) in the paper, I probably would have proposed the same tools that the authors suggest. That being said, I'm not sure it's a bad thing. Sometimes just writing down a problem in a clear way, such that the solution just falls out, is the real contribution. I could believe that to be the case here as well. One other minor critique would be the authors' use of conformal inference for confidence intervals. In the fairness case, it seems like people are likely not going to be happy with the group-level confidence intervals that conformal methods provide. But again, I know nothing about this literature. If the community feels okay with that notion of uncertainty then it's a reasonable method. Still, there should be some (brief) discussion of the difference between frequentist confidence intervals and conformal intervals.

Correctness: Everything seems correct to me.

Clarity: Yes.

Relation to Prior Work: Yes.

Reproducibility: Yes

Additional Feedback:

[Author Response · NeurIPS 2020]

We thank the reviewers and the editor for their helpful comments, which will improve our manuscript. We are pleased to see two positive reviews, and we believe that the most serious concerns raised by the two more critical reviewers are due to easily-addressed misunderstandings; we proceed with a point-by-point response next.

Reviewer 1 (R1) suggested that it will be valuable to see the trade-off between predictive loss and equalized odds violations for varying hyperparameter values. We report on the trade-off in Figure 1 and will include this in the manuscript.

Figure 1: The effect of the regularization parameter for MEPS data set. The results are evaluated on 20 random splits of the data. We use a deep neural network as the underlying predictive model.

Reviewer 2 (R2) expresses concern that the proposed framework does not have a theoretical guarantee. While correct, we are not aware of any learning algorithm that is supported by a guarantee of conditional independence, except in the special case of binary classification. The reason is that this is an extremely difficult problem, especially for neural network algorithms that involve the minimization of non-convex loss functions. Working within these constraints, we rigorously derived our proposed loss function, emerging from the property we proved in Proposition 1. To quote Reviewer 4 (R4), our proposal "uses principled methods and thinks mostly rigorously about the correct way to approach equalized odds." Importantly, for identical levels of fairness, our simulations show the proposed procedure leads to improved accuracy over the alternative heuristic procedures. As a closing remark, because theoretical guarantees are never available, we pay close attention to our new hypothesis test for rigorously detecting violations of the equalized odds property.

R2 also expresses disappointment that our hypothesis test can be viewed as a special case of the abstract Holdout Randomization Test (HRT), as we state in Section 3. We believe this fact does not diminish its utility; the HRT is a special case of the conditional randomization test [18], but why is this a weakness? There is value in being concrete and in proposing novel useful methods: our proposal is the only tool in the literature to rigorously check whether a learned model actually obeys equalized odds. To quote R1, "The [...] general statistical test for equalized odds is also valuable." We are also puzzled about R2's question whether "the hypothesis test is better than the existing notion of equalized odds." In contrast, our test checks whether equalized odds is satisfied by a predictor. This test does not introduce a new notion of fairness. We will clarify this and further clarify that our contribution is twofold: (1) a technique for fitting models that satisfy equalized odds in regression and multi-class classification problems that is shown to be more accurate than the few alternatives and (2) the first rigorous test to detect violations of this property.

Turning to Reviewer 3 (R3), the concern most vigorously expressed is that *fairness through unawareness* has key shortcomings, which we emphatically agree with. To be explicit, *fairness through unawareness* [1] is a notion of fairness where the analyst does not use the sensitive attribute in modeling. In contrast, equalized odds is a different notion of fairness proposed in [6] to address the problems R3 describes. That prior work establishes the benefits of using equalized odds instead of *fairness through unawareness*. A primary goal of our present work is precisely to avoid the problems of *fairness through unawareness* articulated by R3 by introducing a technique to instead achieve equalized odds—a better notion of fairness—in regression and multi-class classification problems. We are therefore puzzled as to where the discussion about *fairness through awareness* is coming from.

Secondly, we must point out an omission from R3's summary that we simply use the randomly sampled sensitive attributes "instead of real sensitive features." Our use of randomization is more sophisticated than it may seem: we use the fair dummies to define a loss function that aims to drive the predictive rule $\hat{f}$ to achieve equalized odds. Certainly, other forms of randomization have been studied before, but randomizing *conditionally* on the observed $Y$ is the crucial idea necessary to promote equalized odds in model fitting, and the technique we provide is entirely new.

Lastly, we thank R3 for asking to explain how to estimate $P(A|Y)$ when $A$ is a continuous sensitive attribute. In short, one can sample fair dummies by fitting a conditional distribution estimator, e.g., using quantile regression, and we will add a proper discussion. In response to R3's related question, equalized odds does not exclude or discourage a perfect predictor. Due to limited space, we must refer the reader to [6] rather than discuss this point.

Finally, R4's most serious concern is about novelty: "I probably would have proposed the same tools that the authors suggest." We believe this comment strengthens the validity of the proposed method and stress that despite the widely-discussed importance of addressing fairness in machine learning, the fundamental problem of fitting models that satisfy equalized odds in multi-class classification and regression problems remained entirely open until recently, and our approach offers improved accuracy over the very few alternatives.

[Meta-Review · NeurIPS 2020]

The introduction of randomization tests for the assessment of fairness is very useful, and the proposed method for encouraging fairness in an adversarial learning system is relevant and novel. The discussion phase showed that the paper would benefit in discussing this work in a broader context, beyond parity measures. First, to shortly describe the pros and cons of addressing fairness through these parity measures (also taking the caution words mentioned in the broader impact section). Second, this would allow to better contrast the randomization of the sensitive attribute that is carried out here with "interventions" on the sensitive features. In particular such as (i) a scheme where randomization would be simply used to mask the sensitive attribute, and (ii) a rudimentary assessment of counterfactual fairness that would be obtained by simply flipping the sensitive attribute. To quote a comment in the discussion: "It has been shown in [Counterfactual fairness, NeurIPS 2017], that intervening sensitive feature is not enough to mitigate unfairness, as there are other features that could be correlated with the sensitive feature (e.g. gender and marital status, race and zip code). So when intervening on the sensitive feature, we have to change also those non-sensitive features. More extensive review: Causal Reasoning for Algorithmic Fairness (in particular section 5.2.3 Equalised Odds and Calibration) by Loftus et al, 2018 https://arxiv.org/pdf/1805.05859.pdf. So in my opinion, there should be at least a discussion on this topic, and ideally a small evaluation on at least one of the datasets where causal graphs have been provided. The causal model for the two most frequently used datasets (Compas and Adult) have been derived: Razieh Nabi and Ilya Shpitser, Fair inference on outcomes, AAAI 2018, Figure 2." In addition, although the paper states early that the approach also applies when features contain (a proxy of) the sensitive attribute (X contains A), this point is not further developed in the paper; a didactic illustration could be useful. check for typos (Prop. 1, L. 121)